# Rapid Evaporative Ionization Mass Spectrometry: A Review on Its Application to the Red Meat Industry with an Australian Context

**DOI:** 10.3390/metabo11030171

**Published:** 2021-03-15

**Authors:** Robert S. Barlow, Adam G. Fitzgerald, Joanne M. Hughes, Kate E. McMillan, Sean C. Moore, Anita L. Sikes, Aarti B. Tobin, Peter J. Watkins

**Affiliations:** 1Agriculture and Food, Commonwealth Scientific and Industrial Research Organization, Coopers Plains, QLD 4108, Australia; Adam.Fitzgerald@csiro.au (A.G.F.); Joanne.Hughes@csiro.au (J.M.H.); Kate.McMillan@csiro.au (K.E.M.); Anita.Sikes@csiro.au (A.L.S.); Aarti.Tobin@csiro.au (A.B.T.); 2Agriculture and Food, Commonwealth Scientific and Industrial Research Organization, Werribee, VIC 3030, Australia; Sean.Moore@csiro.au (S.C.M.); Peter.Watkins@csiro.au (P.J.W.)

**Keywords:** REIMS, red meat, provenance, quality, food safety, lipidomics, metabolomics

## Abstract

The red meat supply chain is a complex network transferring product from producers to consumers in a safe and secure way. There can be times when fragmentation can arise within the supply chain, which could be exploited. This risk needs reduction so that meat products enter the market with the desired attributes. Rapid Evaporative Ionisation Mass Spectrometry (REIMS) is a novel ambient mass spectrometry technique originally developed for rapid and accurate classification of biological tissue which is now being considered for use in a range of additional applications. It has subsequently shown promise for a range of food provenance, quality and safety applications with its ability to conduct ex vivo and in situ analysis. These are regarded as critical characteristics for technologies which can enable real-time decision making in meat processing plants and more broadly throughout the sector. This review presents an overview of the REIMS technology, and its application to the areas of provenance, quality and safety to the red meat industry, particularly in an Australian context.

## 1. Introduction

Australia’s red meat supply chain participants maintain access to global markets through the provision of assurances relating to safety, quality and provenance. There can be times though when fragmentation can arise within the supply chain, creating vulnerabilities which could be exploited. This risk needs to be reduced, if not removed, to ensure that the meat products enter the market with the desired quality, safety and provenance attributes. The associated cost with this risk is difficult to determine; in the Australian context though, fraud is estimated to cost the meat and live animal trade approximately AUD 272 (USD 204) million per year [1] with market non-compliance estimated to cost between an AUD 127 to 163 (USD 95 to 122) million per annum to the beef industry [2] with failures to meet quality parameters and/or food safety requirements as key contributors to this cost.

The meat industry can benefit from innovative technological advancements that can impact all points of the supply chain. For Australia, these can support its status as a provider of clean, green and wholesome red meat products and allow it to remain as a leading global exporter of red meat, shipping approximately AUD 15 (USD 11) billion in red meat and livestock in 2017–2018 to overseas markets [3]. Growing interest also exists from consumers, particularly in markets with strong premiumisation of red meat, for information that relates to provenance, animal welfare, sustainability, quality and safety of these products. The adoption of innovative biological and digital technologies by the red meat sector will assist in verifying the credentials of red meat products. Novel technologies which have multiple application points across the supply chain will therefore be of interest as the industry looks for opportunities to enhance the quality and safety of their products whilst assuring the provenance claims for consumers.

Rapid Evaporative Ionisation Mass Spectrometry (REIMS) is a novel ambient mass spectrometry technique originally developed for use in human medicine for the real-time classification of biological tissue [4]. It has subsequently shown promise for a range of food provenance, quality and safety applications with its ability to conduct ex vivo and in situ analysis. These are regarded as critical characteristics of technologies that can enable real-time decision making in meat processing plants and more broadly through the associated supply chain. This review presents an overview of the REIMS technology, along with a range of different research which are applicable to the areas of provenance, quality and safety and the relevance of such applications to the red meat industry, particularly in an Australian context.

## 2. What Is REIMS?

REIMS is a recent development resulting from the rapidly growing field of advancement in ambient ionisation-mass spectrometry techniques [5]. The REIMS system, produced by the Waters Corporation (Waters, Wilmslow, UK), is the combination of an innovative sampling device, the intelligent knife (iKnife), with a quadrupole time-of-flight mass spectrometer (Q-TOF-MS). When used in tandem, information-rich mass spectral data can be collected in real-time, providing chemical and molecular profiling in seconds via MassLynx MS and LiveID software packages [6]. While Q-TOF mass spectrometers can be regarded as commonplace in laboratories worldwide, the iKnife is a unique handheld sampling device which enables the rapid and direct analysis of intact biological samples [7,8]. The use of this device reduces the need for complex sample extraction and/or preparation techniques that are often required for conventional mass spectrometric assays.

The iKnife was initially developed in the 2000s as a surgical tool to discriminate cancerous cells from healthy cells in situ during surgical procedures [9]. It is an electrosurgical knife powered by a diathermy generator and connected to a mass spectrometer via a length of tubing. The diathermy generator applies a high frequency, alternating current to the iKnife, and a circuit is formed when the iKnife’s electrode contacts the sample’s surface with a return electrode plate placed on or underneath the sample. Upon contact, localised heating cauterises the sample causing the tissue to vapourise and generate a vapour plume consisting of a mixture of aerosolised sample components, water, and positively and negatively charged molecular ions [10]. The vapour is drawn into the attached Teflon tubing using a Venturi air pump device and introduced into the REIMS ion source via a heated transfer capillary. The sampled vapour is infused with isopropanol to aid further ionisation. The charged ions travel through a StepWave^TM^ ion guide to remove contaminants prior to being introduced to the mass spectrometer.

Following the iKnife’s development, additional sampling devices for REIMS have been created to accommodate different sample workflows. For example, bipolar forceps with integrated electrodes in each tip have been developed to remove the need for a separate return electrode [11]. This approach has been applied to classify intact bacteria from an agar plate [12]. A modified, stainless steel pure tip probe manufactured by the Tecan Group has been used to facilitate an automated high-throughput method for classifying clinically important microorganisms [13]. The probe was used with an automated sampling machine to analyse agar plates with minimal user input. Finally, a temperature-controlled soldering iron has been used [14] to sample honey, since an iKnife was not suitable to generate sufficient vapour due to the low conductivity of the honey matrix. All these sampling mechanisms work in a similar fashion and are used to generate a sample vapour for use with REIMS. Further, they demonstrate the flexibility of the technique to adapt to different sample matrices and real-world applications. This of particular importance in the context of food applications where emphasis will be placed on the ability to sample large numbers of samples without the need for extensive cleaning regimes to be implemented.

The REIMS system utilises a Q-TOF-MS which is used to identify the ionised components sampled with the iKnife, or other sampling device. It consists of four key parts: quadrupole, a collision cell, flight tube and detector which characterises the ions by their respective mass to charge ratio (*m*/*z*). After the ions enter the flight tube, they are distributed based on their flight time with the ions with lower *m*/*z* reaching the detector first before the ions with higher *m*/*z* ratios [15]. This produces a spectrum of detected ions based on the *m*/*z* and the related relative abundance. The addition of the quadrupole allows further confirmation since specific ions of interest can be investigated with their selection from the original mass spectrum. The selected ions can then be fragmented in the collision cell which generates a secondary fragmentation pattern, that serving as an additional confirmation step and allowing further molecular identification where required.

The information generated by the REIMS platform creates a unique mass spectral ‘fingerprint’ of the analysed sample that can be used to assess key attributes and differences of, and between, specimens [16]. By taking several cuts of a sample, a semi-quantitative spectrum of key compounds and metabolites can be determined and thus used as biomarkers to predict inherent attributes of the assayed sample. By quantifying the difference in abundances, presence or absence of these biomarkers, predictive modelling built from a library of mass spectral data can be used to indicate aspects of provenance [17], food quality [18], and food fraud [8] often with a high degree of classification accuracy. This entire process from sampling with the iKnife, mass detection, and completed data analysis can be completed in minutes [19] making REIMS amenable to high-throughput analysis. The combination of rapid analysis times with the resolution and accuracy afforded by Q-TOF-MS when coupled with the iKnife makes REIMS an attractive possibility for deployment as a tool for classification across the food industry.

### 2.1. Compounds Amenable to REIMS Analysis

Meat is principally composed of water along with protein and amino acids, lipids (fats and fatty acids) with other minor components in relatively lower proportions [20]. For REIMS, the lipids, and in particular phospholipids, are the most abundant species detected by this technique [11]. These compounds more readily form molecular ions that are detected by the Q-TOF-MS. The protein component of the muscle tissue has not been reported to significantly contribute to the signal measured by the technique. Most likely, this is due to the precipitation of structural proteins in the muscle which results from heating by the iKnife used to generate the plume from the meat for analysis [10]. This does not exclude the possibility that protein and other meat components could not contribute to the REIMS signal since they do so in other food related systems. For example, REIMS has been used to characterise protein expression in bacteria using synthetic proteins [21] and for the identification of adulteration and botanical origin of honey [14]. In the latter case, it was presumed that the compounds detected were Maillard reaction products formed from heating the carbohydrates in honey.

Most recent work relating the application of REIMS to animal muscle reports on the lipid content of the tissue; that is, triacylglycerols, fatty acids and phospholipids. Table 1 summarises recent publications on the use of REIMS with muscle tissue, covering beef and other animals. Only a small number of publications relating to beef and other red meats were available, so a broader approach was adopted to include other animal types to identify any commonality which might exist between different animal muscle groups. Predominantly, it is the fatty acids and phospholipids as a class, along with some other minor components, which are detected by REIMS (Table 1). A range of different ions were reported for these applications with different compounds identified as potential biomarkers for the related work. Ideally, biomarkers which were consistent across for each different application allow a common approach to be used with REIMS. However, this was not the case as different ions were found for each animal species.

The application of REIMS in food falls into two broad areas: (i) food fraud/adulteration (pork and chicken [19] along with fish [22,23,24]) and (ii) tissue classification (meat products [7], beef offal [8], porcine organs [4] and turtle [23]). Other areas of research have been for the screening of β-agonists in pigs [16] and boar taint [18] as well as the characterisation of beef quality [25] and lamb flavour [26]. In relation to beef quality, REIMS was deployed to develop predictive models for sensory classification (tenderness, flavour and juiciness attributes) and tenderness (slice shear force and Warner-Bratzler shear force) while, for lamb flavour, the technique was used to distinguish between lamb, yearling (hogget) and mutton as well as grain versus grass based production systems with the application of different statistical classification techniques.

### 2.2. Data Analysis/Chemometric Approaches

Data analysis is an important consideration with the application of the REIMS technique, as with any metabolomics approach. For REIMS, after collection, the mass spectra are exported from the instrument’s software and the relative content of each peak is normalised so that the major peak abundance is set to 100% with the other peaks in the mass spectrum compared as a ratio to the base peak [27]. The resulting data sets are usually very large and multivariate statistical analyses are needed to analyse the data. As an example of the data sets resulting from REIMS, one was reported to consist of 583 samples with each sample consisting of over 4000 data points [28].

Two different approaches are usually used with data sets. The first is principal component analysis (PCA) which is an unsupervised method where no assumptions are made in relation to the data, and it is used as a visualisation technique to identify any patterns within the data set [29]. The use of PCA for data visualisation has been deployed for most studies [7,16,18,19,22,23,24,27]. As noted above, prior to PCA, the data set is normalized to the base peak, but other scaling techniques can be used as well. These could include centering or scaling the data [29]. For the former, the variable mean is subtracted from the data (i.e., scaled data = $x\;-\;\bar{x}$) while, for the latter, the data can be centered using *z*-scores (i.e., centered data = $\frac{x\;-\;\bar{x}}{\sigma })$ [29]. The second approach is partial least squares discriminant analysis (PLSDA) which is used as a classification technique and assumes that the identity of each grouping is correct [29], and as such can be regarded as a supervised method. PLSDA has been primarily used for classification [7,16,18,19,22,23,24,27]. PLSDA is not the only technique which can be used for classification of REIMS data. Other classification techniques (support vector machine with a linear kernel, radial kernel, and polynomial kernel, random forest, K-nearest neighbour, linear and penalized discriminant analysis, extreme gradient boosting, logistic boosting) along with PLSDA have been compared in one study but mixed results were found with the use of the different techniques [28]. The authors found that not one algorithm could be universally applied for the data set and concluded that that a “one size fits all” approach was not optimal for developing classification models for REIMS data [28].

### 2.3. Comparison of REIMS to Other Ambient Mass Ionisation-Spectroscopy (AMS) Techniques

The development of approaches based on ambient ionisation-mass spectrometry (AMS) have increased in recent times, both in terms of techniques (>90, [30]) as well as the scope of potential application areas. The unique feature of AMS is its ability to directly analyse samples in an open environment at ambient conditions without the usual need for prior chromatographic separation often required for traditional MS analysis. Analysis times are rapid and significantly reduced compared to traditional analytical techniques, allowing high sample throughput, up to 45 samples per minute in some instances [5]. In some applications, the use of AMS techniques have shown comparable results to traditional MS techniques [31,32]. Other MS techniques can be used for applications described in this review; for example, matrix adsorprtion laser desorption ionization (MALDI)-TOF-MS, in identifying the origin of animal meat (pork, beef, horse, veal and chicken) based on protein and gelatin [33] and characterising food components [34]. However, MALDI-TOF-MS is performed under vacuum and regarded out of scope for this review, which has a focus on AMS approaches.

Direct analysis in real-time mass spectrometry (DART-MS) was one of the first AMS techniques developed. A sample can be placed directly in between the DART ion source and the MS interface whereby sample components are desorbed from the sample surface by a stream of heated, metastable carrier gas. Ionisation occurs as the carrier gas reacts with atmospheric components, ionising the amenable, desorbed components before travelling into the mass spectrometer [35]. Desorption electrospray ionisation mass spectrometry (DESI-MS) was also another pioneering AMS technique. In this case, an electrospray emitter produces a spray of charged solvent microdroplets which are directly applied to a sample’s surface with amenable compounds desorbed and ionised prior to introduction into a mass spectrometer [36]. Both DART-MS and DESI-MS can be regarded as the most established of the AMS methods [37] with a diverse range of applications among them, including pesticides in wine [38], melamine and cyanuric acid in milk powder [39], protein identification [40] and phthalate detection in consumer products [41].

Compared to other AMS techniques, there are several key differences regarding sample analysis which differentiate them from REIMS. For REIMS, lipids are the key compounds identified in detection, particularly fatty acids and phospholipids, and are then used as potential biomarkers for classification and predictive modelling. In comparison, both DART-MS and DESI-MS have demonstrated their amenability to a broader range of chemical classes [42,43]. For example, DART-TOF-MS has been used to differentiate between muscle samples from chickens fed a normal diet or a diet containing 5–8% chicken bone meal [44]. By extracting the muscle samples with mixed polarity solvents (water and cyclohexane) with analysis if each fraction, a mixture of sugars, amino acids, organic acids, peptides and lipids could be ionised and potentially used for classification. Similar results were obtained using DART-MS with the same extraction method to analyse carp muscle and differentiate between specimens which had been fed feed supplemented with cereal grains [45]. DESI-MS was used to discriminate between the muscle of beef, pork, horse, chicken and turkey, that had been previously digested with trypsin, by comparing the abundance of various skeletal proteins and peptides [46]. To date, only limited examples have been reported on the use of REIMS for other chemical compounds other than lipids. For example, it has been reported that highly abundant peptides have been ionised in frog skin (*Bombina orientalis*) as well as drugs (lidocaine, gefitinib, cyclosporine A and methotrexate) using REIMS in positive ionisation mode [9]. Other AMS techniques can provide quantitative or semi-quantitative analysis for some applications. DART-MS has been used for quantitating aflatoxin M1 in extracts [47], as well as chemical warfare agents [32], using isotopically labelled internal standards in order to correct for any matrix effect which may either suppress or enhance ion formation. At present, the ability of REIMS to perform quantitative analysis is yet be explored. However, the variability associated with sampling with an iKnife can affect the volume of sample vapour introduced into the MS, along with the challenge of managing any matrix effect due to either ion suppression or enhancement, needs to be addressed before quantitative REIMS analysis is feasible.

The biggest advantage of REIMS is its ability to sample remotely from the MS. This is unique to REIMS compared to other AMS techniques and offers potential benefits for its utilisation outside of laboratory environments and streamlining workflows. With REIMS, the iKnife is connected to the MS by a length of tubing which can be varied in length to suite specific applications. This presents the opportunity for a REIMS system to be deployed with the sampling device and MS separated from each other. For example, in a processing facility, the MS unit can be kept in a “clean” area while an operator uses the iKnife on the processing line as required. This is a distinct advantage of creating a sampling workflow with REIMS compared other AMS techniques.

Unlike REIMS, direct sampling can be problematic with techniques such as DART-MS and DESI-MS since some sample preparation is still often required for their use. For example, direct analysis of a tablet surface using DART-MS fingerprinting to detect falsified anti-malarial medication provided inconsistent results, and the samples needed to be homogenised prior to analysis [48]. DART-MS was used to assess salmon freshness by monitoring muscle lipid abundances over time, and sample homogenisation, extraction and clean up was required prior to the analysis [49]. In contrast (and as noted), REIMS has been demonstrated to be applicable to numerous applications where samples are analysed directly [16,24,50,51] giving it a clear advantage in this regard. The remainder of the review will provide further detail of these applications in the areas of provenance, quality and safety to the red meat industry, with an emphasis on an Australian context.

## 3. Provenance

Consumers are becoming increasingly aware and selective about food provenance and will typically pay for premium product which aligns with their own preferences. The provenance of a food item or ingredient refers to the origin or source from which it comes, and the history of subsequent operations (supply chain) [52]. For the red meat sector, provenance can also relate to the product’s credentials which can be communicated to consumers along the product’s supply chain. Such information could include geographical origin, production system, breed, animal health and sustainability claims. Typically, a product’s provenance is traceable so that the product can be identified anywhere within the supply chain as well as at any point backward or forward within the chain. A valuable feature of such traceability systems is its ability to accurately and rapidly verify product authenticity, species identification and production method of agri-food commodities. Technologies, such as REIMS, which can rapidly verify food product authenticity will likely be components of traceability systems that link the physical and digital attributes of an item across the entire supply chain.

### 3.1. Meat Products

Economically motivated adulteration, or more commonly known as food fraud, is the addition of non-authentic substances to, or substitution of, authentic products for economic gain [53]. Meat species substitution was highlighted in the 2013 European horse meat scandal [54] with animal protein continuing to remain as a target for food fraud. Since then, efforts have increased to develop technologies which can rapidly detect such events. One of the earliest papers highlighting the use of REIMS demonstrated its ability to identify animal tissue with different anatomical origin, breed or species [7]. After conducting a series of experiments aimed at establishing the difference between beef and horse meat samples, the potential of REIMS was evaluated to classify meat patties containing combinations of horse, Wagyu, venison and grain-fed beef. The results showed that it was possible to detect individual meat types at detection limits down to 5% composition in just a few seconds [7]. With cases of food fraud likely to involve product substitution at concentrations much greater than 5%, this was the first successful demonstration of REIMS for rapid lipidomic profiling of meat products. In a similarly themed study, REIMS was utilised for the rapid detection and specific identification of offals within minced beef samples [8]. The study reported that REIMS was able to detect the presence of brain, heart, kidney, large intestine and liver in adulterated beef burgers at concentrations as low as 1%. Furthermore, this could be achieved with raw and boiled samples, although some of the spectral abundances were diminished in cooked products thus indicating that longer cooking times or other cooking types such as grilling or frying may ultimately leave the various offals indistinguishable from each other [8].

In order to use the molecular profiles generated by REIMS to predict the breed, species or tissue types present in a meat sample, classification models must be developed. As previously mentioned, a range of classification systems or machine learning algorithms can be used to develop predictive models with the performance of each algorithm potentially specific to the attribute being evaluated (e.g., production system or breed). For example, a number of different algorithms were evaluated to predict a range of beef attributes including quality grade, production system, breed type, and muscle tenderness and it was found prediction accuracies ranging from 81.5% to 99% were obtainable depending on the algorithm and attribute [28]. It is also feasible that, with the potential for REIMS to sample remotely from the MS, multiple algorithms could be deployed on the data after collection as part of an in-line product assessment system, and allows objective classification and verification of attributes as part of a broader supply chain integrity system.

### 3.2. Fish

As a commodity, fish are commonly substituted and mislabelled on a global basis, with mislabelling occurring in up to 86% of products in some countries [55,56]. Convergent and divergent adaptation creates a significant challenge to morphology-based identification of fish species, with these limitations leading to calls for a molecular approach to fish identification [57]. REIMS has been able to correctly classify (98.99% accuracy) authenticated samples of five different though genetically similar fish species comprising of cod, coley, haddock, pollock and whiting [22]. As part of an expanded investigation, the study identified six samples labelled as haddock clustered with the cod samples. DNA sequencing of the mitochondrial cytochrome c oxidase subunit I gene (COI) confirmed all samples as belonging to *Gadus morhua* species (cod) [22]. In addition to the use of REIMS as a rapid species profiling tool in this study, it was able to separate the haddock samples based on catch method (i.e., trawl or line) presumably through detection of the specific stress metabolites that resulted from the particular catch methods [22].

REIMS has been used to develop predictive models for the classification of economically important species of tuna including bluefin, bigeye, yellowfin and albacore [58]. Models for the classification and distinction of Atlantic salmon and king salmon from rainbow trout were developed following the controversial issuing of a standard to a small group of Chinese companies that attempted to classify rainbow trout as salmon [24]. The substitution of rainbow trout in place of salmon could leave consumers at increased risk of exposure to fish parasites found in rainbow trout potentially resulting in human hepatic disease and cholangiocarcinoma [59]. Rigano et al. [50] developed a major model and three sub models which distinguished 18 marine species typical of the Mediterranean Sea. As well as demonstrating the ability to accurately identify different species of marine animals, the sub models could distinguish closely related species such as horse mackerel and mackerel as well as separate out juveniles from adult greater amberjack [50]. The latter observation suggests that REIMS may be able to distinguish samples based on their age which would have relevance to a variety of agricultural sectors, including red meat.

### 3.3. Botanicals

REIMS has been applied to other food areas and so are included in this review to demonstrate the versatility of this approach, particularly in the area of provenance where it has been used to verify the geographical origin of botanicals. Bronte pistachios are highly prized in global markets due to their intense green colour and aromatic taste that results from growing in the lava land surrounding the volcano, Mt. Etna. REIMS was used to generate spectral data for Sicilian Bronte, Greek, Iranian, Californian and Turkish pistachios and an optimised geographical origin model was subsequently produced [17]. Principal component analysis revealed that several candidate biomarkers that could be used to distinguish Bronte pistachios from the others that were tested [17]. REIMS has also been used for biomarker profiling of the *Kigelia africana* fruit which are of increasing importance for its drug-like properties resulting from numerous secondary metabolites [60]. The study confirmed the presence of antioxidant molecules such as flavonoids and iridoids, and provided insights into the native lipid composition which may contribute to the biological activity of the fruit [60]. Numerous examples of economically motivated fraud have been documented in a wide range of botanicals, particularly herbs and spices [61]. In the near-term, further application areas for REIMS are expected in verifying provenance of botanicals in addition to extensions of models in animal protein sectors with resulting improvements in the associated supply chains.

## 4. Quality

Meat quality is an important consideration for the meat industry as well as consumer acceptance. Presently, many carcass attributes are scored somewhat subjectively by qualified grading experts or assessors and so there is a strong demand from the meat industry for alternative, objective assessment methods not only within Australia, but at an international level as well [62,63]. Objective carcass assessment and on-line assessment is favorable and many approaches are currently being explored [63,64]. As with all ‘omics’ approaches, one novel aspect is that classification does not need to rely on one particular biomarker and biomarkers can be grouped as a profile signature for samples. This is also the case with REIMS, whereby a suite of different biomarkers present in the sample can be used to describe groups or classifications of carcasses. Therefore, although some biomarkers maybe discussed individually, it is the unique profile formed from many biomarkers that can be used to generate classification groups. Some examples of these classification groups are discussed below which are relevant to various meat quality aspects and include animal attributes (e.g., breed or gender), meat attributes, (e.g., tenderness and dark-cutting) or lastly, the treatment of the meat post-mortem (e.g., aging and shelf-life).

### 4.1. Flavour and Sensory Characteristics

In pork boar taint, the main flavor compounds are found in adipose tissue which have been described as having an odour that is faecal-like (indole and 3-methylindole (“skatole”)) or urinary- or sweaty-like (androstenone) and have been successfully classified using an untargeted REIMS approach [18]. In neck fat samples from sows and boars, some specific discrimination compounds were identified in the development of these odours and represent promising targets for an on slaughter-line routine boar taint screening method. Specifically, in the boar taint positive group, the monounsaturated fatty acids (MUFAs) like palmitoleic acid, oleic acid and erucic acid (C16:1, 18:1 and C22:1, respectively), along with saturated fatty acids (SFAs) like lauric acid (C12:0), were predominantly present, whereas in the boar taint negative group, the SFA myristic acid (C14:0) was identified. In addition, the polyunsaturated fatty acids (PUFAs) like linoleic acid and arachidonic acid (C18:2 and C20:4) were also shown to be abundant in the boar taint positive and negative groups. Correlations indicated that several fatty acids were related to androstenone, 3-methylindole and sensory ratings in pork back fat [65,66]. Although, it should be noted that sensory ratings are subjective and can be different between individuals. Therefore, direct correlations between specific fatty acids and sensory aspects may not always be valid. Consequently, care should be taken since “fatty acids are not predictive for the sensory score given by expert panelists who were extensively trained to detect deviant smell in boar fat samples” [65]. Current carcass assessment of boar taint is often performed using a soldering iron, where the neck fat is singed and the released odour is evaluated by a trained assessor. Together, these findings suggest that a more objective approach would be useful for the industry, considering the variations that may exist for a sensory approach, with REIMS offering higher levels of sensitivity and specificity compared to these commonly used sensory methods [18]. REIMS is also practical for such work since measurements can be done quickly (<10 s) and directly on the tissue [67]. Although the initial capital expenditure would be quite large for a meat processor, the high throughput of pork carcasses (600/h in Belgium abattoir) suggests costs per carcass are estimated to remain below 1 € [18].

### 4.2. Breed

In beef, animal breed generates some unique fatty acid profiles which can be related to sensory aspects, such as flavor, and these compounds could be partially responsible for the segregation in clusters between Angus and Wagyu observed in REIMS analysis (Figure 1 [7,25,68]). The unique sensory properties of Wagyu compared with Angus indicate a more intense flavor is recognized, with a higher grilled beef flavor, dairy fat and sweetness attributes, which increase with marbling level, especially in Wagyu beef. These attributes have been related to high levels of MUFA and PUFAs and specific SFAs, like stearic acid (C18:0). Heavy grain fed cattle, like Japanese black cattle, can produce carcasses between 300 to 400 kg which is associated with harder fat having a higher composition of stearic acid compared to other Angus and British breeds [69]. Currently, in the Australian beef industry demand exists for a rapid method that can determine levels of fatty acids to distinguish these characteristics on a carcass. Some preliminary evidence has shown REIMS could be used for breed identification on beef minced meat and *longissimus* muscles [7,28]. Using REIMS with a variety of machine learning algorithm predictive models, Angus have shown a clear separation from other breeds, like Wagyu, where the Angus breed type was predicted with greater than 80% accuracy [28].

This distinctive segregation by breed and associated FA profile would be attractive to geneticists, breeders and primary producers who are interested in attracting a premium price and offering distinct quality segregation and targeted marketing strategies to distinguish their product or brand. In addition, if REIMS could be used to profile and ideally quantify the levels of fatty acids on-line, this would provide an opportunity for meat processors to identify specific vendors or breeding programs for early post-mortem segregations and treatments. For example, carcasses containing lower levels of unsaturated fatty acids (MUFA/PUFA) relative to SFA tend to be associated with harder fat (higher melting point) and thus could be channeled into chillers with a suitable chilling regime appropriate for that type of carcass. Table 2 shows the predominant fatty acids in beef and sheep meat as well as the associated melting points. Optimized chilling could also assist with avoiding issues commonly associated with heavy grain-fed carcasses, e.g., bone taint and repetitive strain injuries from hard fat meat processing. An on-line detection method to help facilitate this segregation process would be beneficial to meat processors, primary producers and associated breeding/marketing programs.

### 4.3. Long Aged Shelf-Life

During long aged shelf-life of vacuum packed chilled beef, there are many volatile and non-volatile metabolite changes which occur that are correlated with the organoleptic sensory perception of the meat [70,71]. These volatile and non-volatile compounds can generate specific signatures that are distinctive over long aged storage (120 to 140 days) and can provide an objective indicator of the quality of product from a consumer perspective. For example, the antioxidant dipeptide carnosine along with lactic acid were among the most abundant non-volatiles and are important attributes of red meat flavor, which both appeared to peak at 84 and 98 days and decreased thereafter. Other non-volatiles such as methionine and tyrosine peaked at 120 days, with fumaric acid and tyramine more prevalent at 140 days. In addition, the concentration of ethanol increased approximately three-fold from 120 to 140 days [70]. Furthermore, the type of packaging can impact on sensory properties and could be explored using the REIMS technology [72]. The impact of the microbial flora during these extended periods on sensory characteristics, and the likely formation of metabolites which could be detected by REIMS need also to be considered [73,74]. These distinctive trends in various compounds over storage life indicate that at each time point, a signature profile would be present that is dependent upon other muscle attributes. Perhaps, with knowledge of animal history and processing regimes, a predictive model could be developed which enables meat processors to make more informed choices about the optimal signature profile that would be suitable for specific customer segments and markets.

Dry aging also generates unique sensory properties of the meat and enhanced beef flavour that could be more easily characterized prior to consumption with the application of REIMS [75]. In beef, the process of dry aging typically generates a more umami and butter-fried flavour that can be preferential to some consumers, compared to traditionally aged products [76]. Perhaps due to the nature of the dry aging process, the applicability of REIMS to determine optimal end point signature profiles, could be more useful from a practical sense, where removal of packaging would not be such an issue.

### 4.4. Nutrition and Essential Fatty Acids (Vitamin E and ω-3)

In recent years, promoting the nutritional benefits of red meat in the diet has been a major focus for the industry, especially in regard to the availability of certain vitamins and essential fatty acids, such as ω-3, with REIMS clearly having the capability for rapid measurement of fatty acids [18]. Nutritionists have focused on improving the ratio of *n*-3 PUFA (formed from α-linolenic acid C18:3) to *n*-6 PUFA (formed from linoleic acid C18:2), due to the associated risks in cancers, coronary heart disease and blood clots leading to heart attack [77,78]. Meat is a rich source of these essential fatty acids and has been well reviewed [78], with “*a 200 g serving of either grass, or grain-fed highly marbled striploin steak providing around 120 mg of ω-3 FAs, making a reasonable contribution to the recommended daily intake for adult males or females (160 and 90 mg, respectively)*” [68,79]. As an example, both Angus and Wagyu beef breeds have shown the total amount of ω-3 FAs, namely α-linolenic, eicosapentaenoic (EPA), docosapentaenoic (DPA) and docosahexaenoic acids (DHA) increase with marbling level [68]. These FAs and other compounds have been linked to positive sensory flavoury attributes and have been well researched but this is outside the scope of this review [78,80,81,82]. Given that REIMS responds to the FA and phospholipid content, this could represent an opportunity to have an online method that could identify different profiles reflective of a brand and thus be used to underpin different marketing strategies for meat producers as well as processors.

### 4.5. Oxidation and Antioxidants

Unsaturated FAs also have a higher propensity to oxidise, which can cause rancidity within a meat product. In addition, oxidation of myoglobin to brown metmyoglobin and the interaction between protein and lipid oxidation are also important aspects of both long aged shelf-life and retail display life [83]. This can have important ramifications on eating quality and consumer perception both in terms of flavor and acceptability. Consequently, the quantity of antioxidants, such as α-tocopherol (vitamin E) and carotenoids are an important attribute of the meat for the industry and can have marketing benefits for grass-feeding and organic beef producers [84]. Supplementation of pasture-fed cattle with α-tocopherol has been shown to be comparable to grain-fed cattle supplemented with supra-nutritional doses of vitamin E [85]. This can modify the yellowness of carcass fat which is regarded as appealing in certain Asian markets, and so be used to create market strategies for meat processors [85]. Current antioxidant analyses are usually performed using high pressure liquid chromatography (HPLC) which is fairly time consuming and laborious. Therefore, if REIMS could be used to quickly measure quantities of antioxidants online, this would represent an attractive tool for meat producers and processors to provide a competitive advantage over some production or feeding systems.

### 4.6. Tenderness

Tenderness is an important meat quality trait, with the consumer prepared to pay more for quality [86]. Meat quality traits, particularly tenderness, depend on both intrinsic as well as extrinsic factors. These factors include: (i) pre-slaughter factors such as species, genotype, nutrition and age of the animal, and pre-slaughter stress; and (ii) post-slaughter factors such as electrical stimulation and hanging of the carcass, ageing of the meat, and packaging and storage conditions [87,88,89]. In Australia, the producer, processor and retailer, are paid more for assuring quality and tenderness under the MSA quality assurance scheme [90]. Therefore, an on-line assessment of tenderness would be valuable to the meat industry, and an opportunity for the deployment of REIMS as well.

As has been noted above, lipids and phospholipids are the most abundant species in biological tissues that readily form molecular ions and are detected by REIMS. The mass spectra can be dominated by protonated and deprotonated intact lipid species, along with the thermal degradation products of these lipids as well. Therefore, many food applications rely on the fact that REIMS spectra are highly specific for different tissues, but it is the distribution of the lipids as groups rather than individual species which is important for differentiation. Many studies presented in the literature are focussed on food fraud applications, such as meat origin (species, tissue type), meat substitution, processing treatments and non-meat ingredient additions [91], which have been discussed elsewhere in this review.

In a series of studies, Montowska and colleagues [46,92] established that AMS techniques (liquid extraction surface analysis (LESA)-MS and DESI-MS) could be used to discriminate between five different cooked meats (beef, chicken, pork, horse and turkey). Subsequently, LESA-MS and targeted MS/MS were used to identify heat-stable peptide markers for each meat type using tryptic digests of raw and cooked meat [93]. Using protein-specific heat stable peptide markers, it was shown that several peptides derived from myofibrillar and sarcoplasmic proteins were resistant to processing, which was validated using a range of meat products [93].

Given that FAs and complex glycerophospholipids are the main species detected using REIMS, which can provide rapid lipidomic profiling of food grade meat products, it has less applicability though for the differentiation of ‘tough’ and ‘tender’ meat based on protein profiles for the reasons previously noted. However, it has been shown that it is possible to identify peptide markers using sample digests and other AMS techniques [93] and that REIMS has been also able to detect amino acids, peptides from aqueous solutions [11]. Therefore, the potential does exist to investigate and validate quality parameters such as tenderness using REIMS using proteins as the analytes.

### 4.7. Meat Colour, pH and Water Holding Capacity

There is some preliminary evidence that suggests REIMS could be used to identify beef dark cutting carcasses, classified under the USDA grading system where lean muscle colour, rather than pH, is used to generate the classification [28,94]. The REIMS analysis used a specialised PLSDA to classify 290 beef *longissimus* muscles into five carcass types, with “*clear separation between classes, with a slight overlap between dark cutter and grass-fed classes*” [28]. This group also overlapped with the grass-fed category and maybe confounded with this effect. In Australia, using 2017–2019 non-compliance figures, grass-fed and non-grain-fed animals are known to have a higher incidence (9.9%) of non-compliance compared with grain-fed animals (2.1%) which is also likely in the US [95]. As described above, the feeding regime and nutritional status of the animal prior to slaughter is a key aspect of determining the lipid composition of the carcass, and therefore it seems likely that the dark-cutting segregation observed in the REIMS study may have been an effect of this. Additionally, during the early post-mortem period, muscles are still in a dynamic phase which develops the structural light scattering properties and can impact on the grade observed [96,97]. Thus, the dark meat observed in this USDA REIMS study may have simply been graded prematurely.

Ideally, meat processors seek a predictive model for non-compliance detection (both high pH and dark meat colour) which would enable the identification of susceptible animals prior to slaughter, or failing this, early in the slaughter floor production line, so that corrective actions can be taken (as found in a previous Australian industry survey [98]). However, considering the REIMS method detects *m*/*z* profiles composed of predominantly lipids, even in medical practices [11], rather than protein, there could be some difficulty in discriminately assessing muscle type causing quality issues such as dark, high pH meat. Furthermore, issues with excessively high or low water holding capacity, like heat-induced toughening and pale, soft, exudative meat, which are also relatively protein-based issues, would likely fall into this category, too. This does represent an opportunity for the exploration, and validation, of REIMS with these protein-based quality issues since, if successful, this would be incredibly useful for the meat industry.

## 5. Safety

Red meat provides an ideal medium for the growth of bacteria which includes both spoilage and food poisoning organisms. In Australia, the Department of Agriculture and Water Resources (DAWR) has a set of standards that export abattoirs must comply with, which includes the National Carcase Microbiology Monitoring Program (NCMMP) which monitors aerobic plate counts, *E. coli* (process control) and *Salmonella* (pathogens). Exporters to the USA and Canada must also test for *E. coli* O157:H7 and six additional Shiga toxigenic *E. coli* (STEC) [99]. These testing regimes come at a significant cost to industry with some testing (STEC and O157:H7) requiring a “test and hold” of meat product. Current methods of microbial testing rely on phenotypic, biochemical and DNA based methods. These methods require technical expertise and labour intensive requiring some tests to take days or weeks to complete, incurring significant costs to industry.

“-Omic” approaches and new technologies are revolutionising the way that microbiological testing is being performed. Since its advent in 1970, MS has been employed in microbial identification schemes with significant progress occurring with the discovery of new ionisation methods, matrix-assisted laser desorption/ionisation (MALDI) and electrospray (ES), in the 1980s along with the development of databases for microbial identification in 2000s. This led the way for the development of commercial systems such as the Bruker MALDI Biotyper and the Biomerieux VITEK MS systems which were granted FDA approval in 2013 and are now routinely used in clinical laboratories for the identification of bacteria, fungi, mycobacteria and resistance markers [11,100]. Despite offering faster turnaround than traditional based methods, MALDI-TOF analysis does require significant sample preparation, whereby the colony is cut from a Petri dish, transferred onto a target plate and placed in a matrix before analysis [101]. This system has proved successful in proteomic profiling, however metabolic profiling cannot be achieved due to lysis of the lipid rich cell wall by the addition of a matrix. Metabolomics is a growing field and it is now widely accepted this ‘-omics’ approaches provide information much closer to the phenotype than other DNA based methods, or other ‘-omics’ technologies. The application areas for REIMS are broad and varied, and potentially could help overcome some of the challenges faced by the red meat industry.

### 5.1. REIMS Applications in Bacterial and Yeast Speciation

The bacterial cell wall is lipid rich and lipid configurations are very much species dependent. This is where the REIMS technology is advantageous, providing a mass spectrum rich in lipids and metabolites. Employing this principle, Strittmatter and colleagues [12] carried out a proof of concept (POC) study on nine clinically relevant Gram-positive and Gram-negative bacterial species using REIMS. For ease of sampling, the iKnife was replaced with a pair of bipolar forceps (Figure 2A) to allow direct sampling of a colony grown overnight on bacteriological agar. Squeezing the forceps together triggers an electrical current which heats the cell until it bursts forcing a lipid rich vapour to travel through a small hole, along tubing to the REIMS mass spectrometry unit (Figure 2B).

**Figure 2 metabolites-11-00171-f002:**
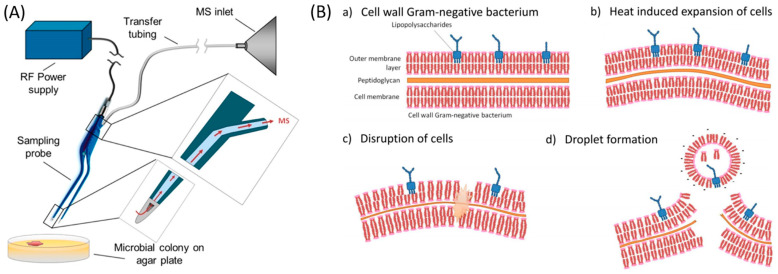
(**A**) Experimental setup for bacterial analysis using bipolar forceps with REIMS (reproduced from [12]), and (**B**) droplet formation in REIMS (reproduced from [102]).

The mass spectrum produced was in the range of *m*/*z* = 150–2000 for most species with a dominance of phospholipids in the mass range of *m*/*z* = 600–900. Statistical modelling of the data output showed clear differentiation of Gram-positive and Gram-negative bacteria. Whist both groups showed the presence of phosphatidylglycerols (PGs), di-phosphatidylglycerols (DPGs) and, to a lesser extent, phosphatidic acid (PA), only Gram-positive bacteria produced phosphatidylethanolamine (PE) signals. It is known that bacterial cell walls and cell membrane are mostly made of PGs, PEs and DPGs and that Gram-positive bacteria contain more PE which correlates with these findings. The background contribution of lipids resulting from the five bacteriological agars used to culture the bacteria were assessed by comparing mass spectra readings with no signals detected from any of the five agars [103]. Multivariate statistical analysis of the data showed REIMS was able to identify species with 95.9% accuracy, genus with 98.7% and gram stain with 100% accuracy. This study also incorporated a POC for yeast identification and successfully identified five pathogenic *Candida* yeast species (fungi) with 98.8% accuracy [103]. Fungal identification by MALDI-TOF has been limited as, unlike REIMS, it requires extensive sample preparation. An extensive investigation into the use of REIMS to identify *Candida* isolates correctly classified 153 isolates to species level with 100% accuracy [104]. 

One of the most attractive features of the REIMS is the lack of sample preparation time and the speed of analysis. Bolt et al. [13] further developed the pace at which the REIMS can be operated by the addition of a robot which they termed “high throughput (HTP) REIMS” making it ideal for incorporation into semi-automated commercial or clinical laboratories. The robot was made with the Tecan Freedom Evo robotic workstation with the addition of the Scirobotics Pickolo colony-picker which features a robotic arm for grabbing the agar plate and placing it on an imaging stage. A second robotic arm samples the microbial biomass with a probe with colony selection made by either the robot’s software or the user. Rapid sample turnaround is ideal for the meat industry given the short-shelf life of meat, and cost of storage required pending release of product following microbiological analysis. This robot can process 3000 to 4000 colonies in a 24-h period with minimal input from the user [13]. 

As well as providing in situ (ambient) testing, REIMS provides the novel ability to also test samples ex vivo for example human biofluids and tissue or food matrices. Traditional methods such as HPLC and gas chromatography (GC) coupled with MS have been used to test human faeces with both techniques requiring extensive sample preparation. Direct sample metabolomic analysis of faecal material was achieved by REIMS following modification of HTP REIMS to incorporate a robotic tip with a monopolar electrode and the use of a Petri dish coated in conductive material [105]. The faecal material was placed on a Petri dish on the imaging platform coated in a conductive material to act as a return electrode. The REIMS unit strongly detected bacterial phospholipids PG and PE amongst other constitutes of the faecal matter such as bile acids and highlighted the possibility that REIMS could be used to identify pathogenic bacteria direct from samples using these different lipid profiles as biomarkers [105]. Further work is required to determine the feasibility of this approach and what level of taxonomic classification could be achieved. Whilst this work was carried out on human faecal samples, it has potential to be applied to cattle faeces which could help in our understanding of the effects of prebiotic or probiotic treatments, the effect of diet and other environmental factors on the presence and shedding of bacterial pathogens.

### 5.2. Subspecies Differentiation

A POC study was used to assess REIMS’ ability to serotype a small set of 10 strains consisting of three serotypes of *Streptococcus pneumoniae* [102]. There are pitfalls to traditional serotyping methods such as genomic methods (PCR) which are laborious and antigen-antibody serotyping, which is equally time consuming, lacks sensitivity and is expensive. As is the case with foodborne pathogens such as STEC, there are many serotypes of *S. pneumoniae*, yet only a small subgroup of these serovars are implicated in human illness. The classification of serotypes by REIMS offered 77.8% accuracy in this study with the conclusion that adjustments were needed to the classification model and a greater pool of serotypes and strains were required to increase this accuracy. However, this study demonstrated the feasibility of serotyping using REIMS [102]. In a subsequent study, the potential of REIMS to differentiate seven *E. coli* isolates, consisting of six derivatives of laboratory strain K12 and one of laboratory strain B, was assessed. REIMS differentiated these strains with an accuracy of 87.3% (Figure 3) with misclassification confined to two K12 derivatives; MC1000 (red) and MC4100 (yellow). Once again, the study concluded that the classification accuracy would be enhanced by the addition of larger numbers of isolates [102]. Sub-species differentiation methods have traditionally relied on genomic techniques such as pulsed field gel electrophoresis (PFGE), multi-locus sequence typing (MLST) and WGS (whole genome sequencing). The use of REIMS to differentiate the MLST type of 45 *Pseudomonas aeruginosa* isolates was evaluated and found to be able to accurately predict MLST type with an accuracy of 83% [106]. These studies show metabolic fingerprinting by REIMS can be used to determine some level of sub-species classification [12,103,107].

### 5.3. Quorum Sensing Molecules—Virulence and Antimicrobial Resistance 

Traditionally, virulence and anti-microbial resistance (AMR) prediction has been performed using a variety of genomic techniques that determine the presence of virulence and AMR genes. Whilst evaluating the ability of REIMS for MLST prediction, Bardin and colleagues [106] noted that along with a dominance of phospholipids, the mass spectral output also highlighted the presence of 17 rhamnolipids and 18 quorum sensing molecules (QSMs) were also seen (Figure 4). *Pseudomonas*, like *E. coli*, uses QSMs to control the expression of virulence genes [107]. This study compared the QSMs of *P. aeruginosa*, which had been implicated in human illness, with non-pathogenic isolates. How the isolates were grown (i.e., as single colonies or a lawn) produced differences in the QSM detected. Furthermore, there was a correlation in the detection of some QSM with the presence of cell wall phospholipids, suggesting QSM have some influence on structural modifications [106]. The ability of REIMS to detect QSMs shows REIMS can be used for metabolite fingerprinting to assess virulence potential.

AMR is of importance to the red meat industry globally as well as in Australia. The US Center for Disease Control (CDC) considers strains of multidrug resistant *Salmonella* to pose a serious food safety threat. Proposals have been made to urge the US Food Safety and Inspection Service to declare these strains as adulterants in meat and poultry products [108]. Antimicrobial testing is also important for antimicrobial stewardship; the Australian Lot Feeders’ Association (ALFA) have antimicrobial stewardship guidelines which the Australian feedlot industry need to adhere to. AMR monitoring and surveillance is a key measure of the successful implementation of an antimicrobial stewardship program [109]. Current test methods for AMR detection, which include Biomerieux E-test, disc diffusion and commercial systems Phoenix (BD Diagnostic systems) or the Biomerieux Vitek 2, determine the minimum inhibitory antimicrobial concentration by monitoring bacterial growth. These methods are multi step and timely, making the “test and hold” approach that is used with STEC impractical for *Salmonella* testing. Technical issues can also arise for polypeptide antibiotics, anaerobic and slow growing or fastidious bacteria. The application of REIMS to measure AMR was evaluated by assessing the response of *K. pneumoniae* to a carbapenem antimicrobial. Twenty-two strains (10 sensitive and 12 resistant) of *K. pneumoniae* were grown overnight on ISO sensitest agar impregnated with an antibiotic disk containing 10 µg of ertapenem [102]. REIMS was used to take two samples, one close to the disk and the other opposite to the disk. Using multivariate statistical analysis, the predictive model was able to identify resistant or sensitive strain with 75% accuracy with misclassification shown not to be associated with any particular resistance mechanism [102]. Further studies with increased strains and resistance mechanisms may reveal a range of applications for animal and human health surveillance activities.

### 5.4. Chemical Residues

The Australian DAWR implements and administers the National Residue Survey (NRS) where a random as well as targeted approach is deployed to monitor chemical residues in agricultural produce such as meat, egg, honey and aquatic food products. These approaches ensure that the related industries comply with the expected standards that satisfy domestic and export needs as well as international regulation covered by, for example, Codex Alimentarius. The numbers and types of samples used for testing depends upon the produce, related systems and types of chemicals used in each case (e.g., pesticides, veterinary medicines, environmental pollutants, etc.). As part of the targeted residue monitoring program, specific on-farm audits are performed to ensure that these industries are able to comply with the supply of product to both EU and non-EU markets that are particularly sensitive to the use/presence of growth hormones (https://www.agriculture.gov.au/ag-farm-food/food/nrs/animal-residue-monitoring, accessed 28 February 2020). This need for continual monitoring to meet legislative directives requires analytical methods which detect and quantitate these compounds and/or their metabolites.

The two common methods employed for the analysis of chemical residues in food products are HPLC and GC with MS for detection and quantitation. Both techniques employ often long and complex extraction steps to prepare the sample ready for analysis. This has been recognised as a critical step with effort being directed to reduce sample preparation for residue testing. Two examples of this are QuEChERS (quick, easy, cheap, effective, rugged and safe; [110]) and ‘point and shoot’ [111]. These approaches have been deployed with high resolution MS to detect a range of hormones and their metabolites in several types of meats. The study determined a limit of quantitation of 1.0 µg/kg for most of the compounds and subsequently identified the hormones progesterone and hydrocortisone in a beef and pork sample [112]. A similar approach was used to detect and analyse 17 steroid hormones in different sex and maturity stages of Antarctic krill (*Euphausia superba Dana*) with limits of quantitation ranging from 2–100 ng/kg [113].

The application of REIMS for rapid chemical residue testing has yet to be evaluated, and caution will be needed in evaluating its use for chemical residue testing. The expected range of any residue are quite low to ensure product compliance, which means the measured levels will be at least at, or below, the limit of quantitation. It is very unlikely that REIMS will able to detect analytes at this concentration range. As noted in this review, the response from any compound in REIMS principally arises from the FA and PL content that is present in much higher content compared to that of any expected residue (% vs. µg/kg). There is some evidence to suggest that this is the case. Verplankan et al. [18] reported that, using traditional analytical methods for boar taint, that 3-methylindole and androsterone were present in pork at concentrations between 200 and 500 µg kg^−1^, respectively. However, the response from the ions associated with indole, 3-methylindole and androsterone were not found in the mass spectrum and below the signal from the background matrix. Thus, any response was most likely due to the changes resulting from mass spectral differences from the FAs and PLs, and not those from the compounds reasonable for boar taint. As previously mentioned, REIMS is an AMS technique. Direct analysis in real-time mass spectrometry (DART-MS) is also an AMS technique which been used as part of a multi-component solution for the detection of pesticides and adulterants in wines [38] and pyrotechnic residues in seized postal packages containing fireworks [114]. The use of AMS in these applications suggest that REIMS could have utility in the residue testing space enhanced by the capacity to conduct real-time sampling away from the MS, however further work is needed to substantiate this.

## 6. Summary

Australia has a reputation for producing clean, green and wholesome red meat products resulting from a long history of supplying compliant meat to global markets. However, there is increased emphasis from consumers and trade partners to verify product claims in order to achieve premiumisation of red meat products. If Australian red meat producers are to meet the emerging global demands, then a range of digital and biological verification technologies are required. REIMS has potential to rapidly verify or classify red meat products based on a range of provenance, quality and safety attributes. Importantly, the remote sampling benefit afforded using the iKnife or other fit-for-purpose probes would enable REIMS systems to be installed in all food processing environments. Automation of the system would appear to be a natural evolution with ‘off the shelf’ robotics and sensing/visualisation equipment able to transition REIMS into a real-time automated compliance system. POC studies will identify which product claims are of greatest interest to the Australian red meat industry and to what extent REIMS provides a convenient, rapid solution for the industry.

## Figures and Tables

**Figure 1 metabolites-11-00171-f001:**
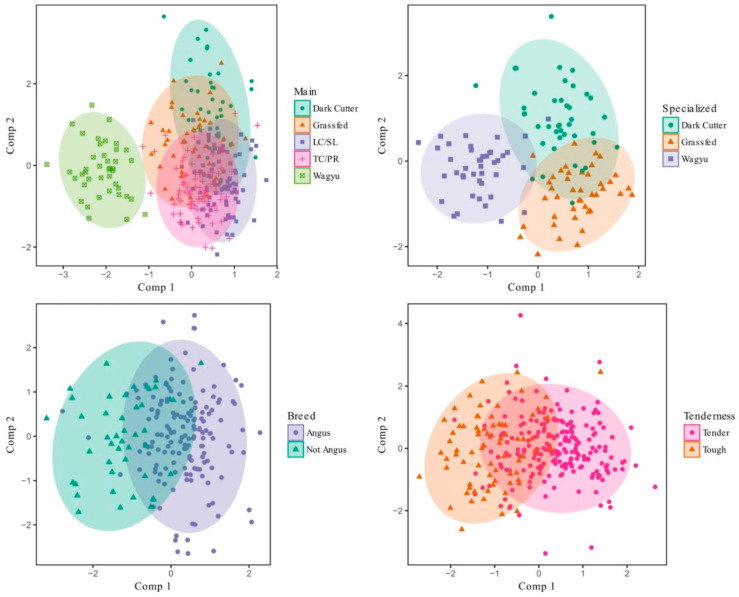
Visualization of the PLSDA model for a variety of model sets relevant to various carcass groups, reproduced from [28].

**Figure 3 metabolites-11-00171-f003:**
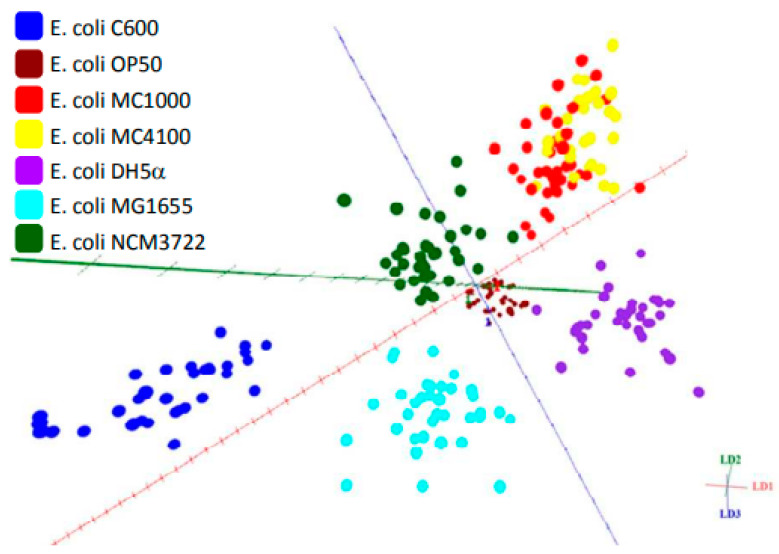
Principal component analysis (PCA)-linear discriminant analysis (LDA) plot differentiating seven strains of *E. coli* using REIMS, reproduced from [102].

**Figure 4 metabolites-11-00171-f004:**
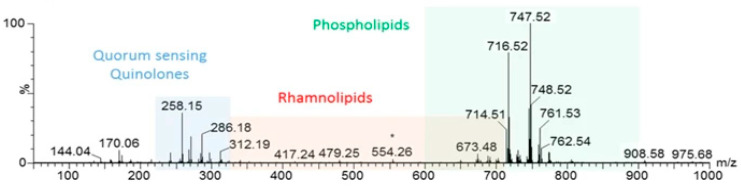
*P. aeruginosa* REIMS mass spectral output showing the presence of typical phospholipids and the additional rhamnolipid and quorum sensing molecules, reproduced from [106]. The * indicates the peak of the leucine enkaphalin which is used a mass standard for REIMS.

**Table 1 metabolites-11-00171-t001:** Chemical species identification in the application of REIMS to animal tissue.

Species	Compound ^A^	Reference
Beef	FAs, PLs	[7]
	Cer, DAG, GluCer, MGDG, PL	[8]
Pig	FAs, PLs	[16]
	FAs, PLs, others	[4]
	FAs, PLs, others	[18]
Pork/chicken	PLs	[19]
Turtle	PLs	[23]
Fish	Lipids—PLs	[27]
	Lipids—PLs	[24]
	FAs, PLs	[24]
	PLs (FA as a dimer)	[22]

^A^ FA = fatty acid, PL = phospholipid, Cer = ceramide, DAG = diacylglycerol, GluCer = glucosylceramides, MGDG = monogalactosyldiacylglycerols.

**Table 2 metabolites-11-00171-t002:** Major fatty acids (% *w*/*w*) in cattle and sheep ^A^.

Fatty Acid	Class ^B^	Beef	Sheep	Melting Point (°C)
Myristic (C14:0)	SFA	2–4	2.5–4	53
Palmitic (C16:0)	SFA	22–28	22–27	63
Palmitoleic (C16:1)	MUFA	1–12	1–2	0
Stearic (C18:0)	SFA	4–30	17–30	70
*trans*-Vaccenic (C18:1)	MUFA	1–12	0.3–4	45
Oleic (C18:1)	MUFA	35–50	19–31	16
Linoleic (C18:2)	PUFA	1–2	2–4	−9

^A^ reproduced from [69] ^B^ SFA = saturated fatty acid MUFA = monounsaturated fatty acid PUFA = polyunsaturated fatty acid.

## Data Availability

Not applicable.

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
