# Peer review of "Rapid Evaporative Ionization Mass Spectrometry: A Review on Its Application to the Red Meat Industry with an Australian Context"

_metabolites, 2021, doi:10.3390/metabo11030171_

Round 1

Reviewer 1 Report

This is a timely, interesting and well-structured paper. I do have some minor comments that should be addressed prior to publication:

Abstract:

  • Line 21: Is the “ability to conduct in vivo analysis” correct in the context of food analysis? Since it is a rather destructive method, it can for example not be applied on live animals. ‘Ex vivo’ may be more appropriate. Due to the focus on real-time sampling and analysis, it should be clear to readers that this technique cannot be applied on live animals.

  1. Introduction
  • Sentences at the beginning of the introduction and the abstract are very similar. Please provide some variation and avoid self-plagiarism. Same comment for line 53-55.
  • Line 37: “with failures to meet quality parameters and/or food safety requirements as key contributors to this cost.” Do the authors have a reference to support this?

  1. What is REIMS (please note that the title has the wrong layout/level)
  • Line 67: can the authors be a little more specific about “modelling and visualisation software”? This is a bit vague.
  • Line 69: the iKnife being ‘novel’ is debatable since development dates back 10 years. The application in food analysis is novel though.
  • Line 86-98: Fairly recently, also a ‘fat probe’ was developed for sampling of fatty tissue. In the context of meat (and fat) this may be relevant to mention as well.
  • Line 180: does the vendor software offer/include PLSDA modelling? Or is this performed using other software programs or approaches? What about LDA modelling? The authors should make a distinction between ‘vendor’ data processing and other approaches being used. On line 172, the authors mention “analyse the data using a vendor specific package and workflow”, but real-time sample recognition with LiveID (vendor) software is based on PCA and LDA modeling. This may be misinterpreted by readers.
  • I am a bit in doubt about the added value of figure 1 as it displays the ‘classical’ metabolomics workflow, but not the REIMS workflow. It would be interesting to see the difference between the two.
  • Line 197-198: ‘usual need’ and ‘often required’ points out the same thing twice?
  • Line 240-244. Do the authors have a reference to support this? Same for line 245-255.
  • Suggestion to include some more information/emphasize the need for the development and use of fit-for-purposes probes besides the iKnife and/or extensive cleaning, since when using the iKnife for food analysis (for a lot of samples), debris builds up easily… Therefore, specific probes and/or extensive cleaning are required, which is not really mentioned at the time.

  1. Provenance
  • Line 280-282: “will be key components”; this is a strong statement, do the authors have a reference to support this or provide more argumentation?

  1. Quality
  • Subtitle: ‘Flavour and sensory’: do the authors mean ‘sensory characteristics’ or maybe sensory evaluation’ or …?
  • Line 390-391: ‘described’ in in there twice.
  • The resolution of figure 4 is insufficient.
  • Line 500-517: suggestion to make a separate section on oxidation/antioxidants. The 4.4. subtitle does not really cover this?

  1. Safety

/

  1. Summary
  • Line 788: only the iKnife is mentioned, maybe ‘or other fit-for-purpose probes’ should be added.

Author Response

Reviewer 1

This is a timely, interesting and well-structured paper. I do have some minor comments that should be addressed prior to publication:

                Response: We would like to thank the reviewer for providing comments on the manuscript and for recognising that it is timely and interesting. We have attempted to address each issue raised and have provided details on what we have done against each item.

Abstract: Line 21: Is the “ability to conduct in vivo analysis” correct in the context of food analysis? Since it is a rather destructive method, it can for example not be applied on live animals. ‘Ex vivo’ may be more appropriate. Due to the focus on real-time sampling and analysis, it should be clear to readers that this technique cannot be applied on live animals.

                Response: The authors agree with the reviewer and have changed in-vivo to ex-vivo in the abstract and at two other points in the manuscript.

 Introduction: Sentences at the beginning of the introduction and the abstract are very similar. Please provide some variation and avoid self-plagiarism. Same comment for line 53-55.

                Response: We thanks the reviewer for raising this point. The first line of the introduction has been modified to read ‘Australia’s red meat supply chain participants maintain access to global markets through the provision of assurances relating to safety, quality and provenance’. Line 53-55 remains but the abstract has been updated to read ‘Rapid Evaporative Ionisation Mass Spectrometry (REIMS) is a novel ambient mass spectrometry technique originally developed for rapid and accurate classification of biological tissue which is now being considered for use in a range of additional applications’.

Line 37: “with failures to meet quality parameters and/or food safety requirements as key contributors to this cost.” Do the authors have a reference to support this?

                Response: These types of costs are routinely met by supply chain participants who, for reasons that are well understood, prefer not to discuss them openly. Therefore, although it’s accepted that costs are incurred, a reference detailing them further is not available. The manuscript does acknowledge this by stating ‘the associated cost with this risk is difficult to determine’.

What is REIMS (please note that the title has the wrong layout/level)

                Response: The manuscript has been checked for consistency with the instructions to authors.

Line 67: can the authors be a little more specific about “modelling and visualisation software”? This is a bit vague.

                Response: The authors have clarified the modelling and visualisation software packages used and have modified the manuscript to read ‘When used in tandem, information-rich mass spectral data can be collected in real-time, providing chemical and molecular profiling in seconds via MassLynx MS and LiveID  software packages’.

Line 69: the iKnife being ‘novel’ is debatable since development dates back 10 years. The application in food analysis is novel though.

                Response: The authors agree with the reviewer’s assessment and have removed the word novel from the sentence.

Line 86-98: Fairly recently, also a ‘fat probe’ was developed for sampling of fatty tissue. In the context of meat (and fat) this may be relevant to mention as well.

                Response: The authors are aware of several developments in sampling devices for REIMS. As many of these are in prototype stage, they are not easily able to be referenced. We have not been able to identify the specific probe mentioned by the reviewer but believe Line 86-98 adequately demonstrates the variability of probes being used and acknowledges the likelihood that more will be developed for specific applications.

Line 180: does the vendor software offer/include PLSDA modelling? Or is this performed using other software programs or approaches? What about LDA modelling? The authors should make a distinction between ‘vendor’ data processing and other approaches being used. On line 172, the authors mention “analyse the data using a vendor specific package and workflow”, but real-time sample recognition with LiveID (vendor) software is based on PCA and LDA modeling. This may be misinterpreted by readers.

                Response: The authors agree that it may be possible for readers to misinterpret these sentences and assume that vendor specific software provides all analytical options/workflows. The reference to vendor specific package and workflow has been removed.

I am a bit in doubt about the added value of figure 1 as it displays the ‘classical’ metabolomics workflow, but not the REIMS workflow. It would be interesting to see the difference between the two.

                Response: The authors considered the value of figure 1 and have chosen to remove it from the manuscript. The emphasis of figure and Section 2.2 was to detail the importance of data analysis and to highlight the variability in approach that can be taken. Reviewer 2 also made a comment relating to Section 2.2 and the section has been updated to address both reviewer’s comments.

Line 197-198: ‘usual need’ and ‘often required’ points out the same thing twice?

                Response: The authors agree and the line ‘This also means that extensive sample preparation is no longer required’ has been removed from the manuscript.

Line 240-244. Do the authors have a reference to support this? Same for line 245-255.

                Response: The authors have removed the generalised statement relating to alternative AMS techniques and the paragraph now focuses on the remote sampling opportunity that REIMS offers.

Suggestion to include some more information/emphasize the need for the development and use of fit-for-purposes probes besides the iKnife and/or extensive cleaning, since when using the iKnife for food analysis (for a lot of samples), debris builds up easily… Therefore, specific probes and/or extensive cleaning are required, which is not really mentioned at the time.

                Response: The authors agree with the suggestion and the following sentence ‘This of particular importance in the context of food applications where emphasis will be placed on the ability to sample large numbers of samples without the need for extensive cleaning regimes to be implemented’ has been added to the paragraph (lines 88-98) that discusses probe development.

 Provenance: Line 280-282: “will be key components”; this is a strong statement, do the authors have a reference to support this or provide more argumentation?

                Response: The sentence has been modified and now suggests technologies such as REIMS will likely be components of traceability systems that link the physical and digital attributes of an item. The Australian red meat industry is conducting research into how isotopes, geochemistry, genomics and technologies like REIMS may be applied in a red meat traceability system. Whilst the digital and data systems will do the majority of work in the traceability space, there is perceived value in being able to biologically verify a product claim.

Quality: Subtitle: ‘Flavour and sensory’: do the authors mean ‘sensory characteristics’ or maybe sensory evaluation’ or …?

                Response: The subtitle has been modified and now reads ‘Flavour and sensory characteristics’

Line 390-391: ‘described’ in in there twice.

                Response: Thanks to the reviewer for pointing this out. The sentence has been modified so that described is only used once.

The resolution of figure 4 is insufficient.

                Response: The resolution of Figure 4 (now Figure 1) has been enhanced and should now meet the requirements of the journal.

Line 500-517: suggestion to make a separate section on oxidation/antioxidants. The 4.4. subtitle does not really cover this?

                Response: The authors agree with the reviewer and have created a separate section ‘oxidation and antioxidants’.

Summary: Line 788: only the iKnife is mentioned, maybe ‘or other fit-for-purpose probes’ should be added.

                Response: The reviewer’s suggestion of adding ‘or other fit-for-purpose probes’ to the sentence has been adopted and the manuscript modified accordingly.

Reviewer 2 Report

The review is dealing with the application of new mass spectrometry technique (Rapid Evaporative Ionisation Mass Spectrometry, REIMS) for analysis of red meat. I see the biggest weakness in poorly chosen figures. Showing a PCA/PLSDA/LDA plot without more detailed descriptions is meaningless. It would be more appropriate to replace these figures with a table or spectrum. In this form it is more like review of statistical evaluation of mass spectrometric data than the application of REIMS and it is pity because the article has high potential.

Line 35-36 - For me, as foreign reader, will be better to get the cost of meat in USD $ or EUR.

Line 45 – It is USD $ or AU $?

Line 61 – Why is this title italic? Other titles are bold.

Chapter 2.1. – The most articles relating the application of REIMS to animal muscle is focusing on lipid content. Authors also saying that protein component of the muscle tissue has not been reported to significantly contribute to the signal measured by the technique. The common mass spectrometric technique for direct analysis of fat in soft tissues is matrix assisted laser desorption/ionization mass spectrometry. It will be beneficial to discuss the pros and cons of REIMS and MALDI in chapter 2.3. where comparison of REIMS to other AMS technique is discussed.

Lines 176-180 – The PCA is powerful tool for evaluation of large dataset. However, the dataset should be transformed, especially for metabolomic study. Please add 1-3 sentence dealing with this data treatment.

Line 180 – If you use the word “unsupervised” for PCA, you should use “supervised” when you describe PLS-DA.

Figure 2 – The figure is in poor quality. Moreover, this figure did not show anything about detection limits down to 5%. If authors would like to use this linear discriminant analysis plot (supervised method), the better explanation should be mentioned in text.

The figure caption of Figure 2 should be rewritten. This figure only shows the variability among groups. Detection of compounds was performed using appropriate analytical instrumentation.

Figure 3- The axis of score plot are not visible. OPLS-DA analysis, as supervised method, is first step for finding differences (markers) in data (next steps are S-plot, Jackknife plot etc.). In my opinion this is not appropriate graph to show in this review. Moreover, without any explanation.

Chapter 3.3. Although this is an interesting chapter, it does not fit into the concept of the whole article. It will be better to remove it, or move it in introduction.

Lines 393-401: It will be very beneficial to show these data in summarizing table, if it is possible.

Figure 4 -  It is illegible.

There are two font size (chapter 4 is written with smaller font size).

Author Response

Reviewer 2

The review is dealing with the application of new mass spectrometry technique (Rapid Evaporative Ionisation Mass Spectrometry, REIMS) for analysis of red meat. I see the biggest weakness in poorly chosen figures. Showing a PCA/PLSDA/LDA plot without more detailed descriptions is meaningless. It would be more appropriate to replace these figures with a table or spectrum. In this form it is more like review of statistical evaluation of mass spectrometric data than the application of REIMS and it is pity because the article has high potential.

                Response: We thank Reviewer 2 for providing comments on the manuscript and confirming the high potential of this review article. We agree with the reviewer’s comments relating to the choice of figures. Research publications in this area typically contain numerous figures which capture the outputs of workflows and the manipulation of data. It is not possible to display numerous figures in this review and therefore have made the decision to remove a number of figures from the manuscript and believe that readers of the review can gain an insight to the application area via the description provided in text and can seek out the original paper should they require further detail. Reviewer 2 also provided several specific comments and we have attempted to address each issue raised and have provided details on what we have done against each item.

Line 35-36 - For me, as foreign reader, will be better to get the cost of meat in USD $ or EUR.

                Response: The authors support showing the cost of meat in USD and have included the USD amounts in parentheses. A conversion rate of 1 AUD to 0.75 USD was applied.

Line 45 – It is USD $ or AU $?

                Response: We have provided clarification that the amounts are AUD and have provided USD equivalents as per the previous reviewer response.

Line 61 – Why is this title italic? Other titles are bold.

                Response: We have conducted a check of all main section headings for consistency with the instruction to Authors.

Chapter 2.1. – The most articles relating the application of REIMS to animal muscle is focusing on lipid content. Authors also saying that protein component of the muscle tissue has not been reported to significantly contribute to the signal measured by the technique. The common mass spectrometric technique for direct analysis of fat in soft tissues is matrix assisted laser desorption/ionization mass spectrometry. It will be beneficial to discuss the pros and cons of REIMS and MALDI in chapter 2.3. where comparison of REIMS to other AMS technique is discussed.

                Response: We thank the reviewer for an opportunity to clarify the scope of this section as it pertains to the overall review. The following sentences “Other MS techniques can be used for applications described in this review; for example matrix adsorption laser desorption ionization (MALDI)-TOF-MS, in identifying the origin of animal meat (pork, beef, horse, veal and chicken) based on protein and gelatin and characterising food components. However, MALDI-TOF-MS is performed under vacuum and regarded out of scope for this review, which has a focus on AMS approaches” have been added to address the reviewer’s comment.

Lines 176-180 – The PCA is powerful tool for evaluation of large dataset. However, the dataset should be transformed, especially for metabolomic study. Please add 1-3 sentence dealing with this data treatment.

                Response:  Section 2.2 has been updated to include the following detail relating to data treatments. As noted above, prior to PCA, the data set is normalized to the base peak, but other scaling techniques can be used as well. These could include centering or scaling the data [30]. For the former, the variable mean is subtracted from the data (i.e. scaled data = ) while, for the latter, the data can be centered using z-scores (i.e. centered data =

Line 180 – If you use the word “unsupervised” for PCA, you should use “supervised” when you describe PLS-DA.

                Response: The sentence describing PLSDA has been updated to include “and as such can be regarded as a supervised method”.

Figure 2 – The figure is in poor quality. Moreover, this figure did not show anything about detection limits down to 5%. If authors would like to use this linear discriminant analysis plot (supervised method), the better explanation should be mentioned in text. The figure caption of Figure 2 should be rewritten. This figure only shows the variability among groups. Detection of compounds was performed using appropriate analytical instrumentation.

                Response:  The authors support the reviewer’s comment and as a result Figure 2 has been removed from the manuscript. The study is adequately described and referenced within the text (lines 277-283).

Figure 3- The axis of score plot are not visible. OPLS-DA analysis, as supervised method, is first step for finding differences (markers) in data (next steps are S-plot, Jackknife plot etc.). In my opinion this is not appropriate graph to show in this review. Moreover, without any explanation.

                Response: The authors support the reviewer’s comment and as a result Figure 3 has been removed from the manuscript. The study is adequately described and referenced within the text (lines 313-316).

Chapter 3.3. Although this is an interesting chapter, it does not fit into the concept of the whole article. It will be better to remove it, or move it in introduction.

                Response: The authors request that section 3.3 remains in the manuscript in its current place. The authors have modified line 334 to remove the words ‘out of scope’. Whilst it is easy to consider provenance purely from the animal perspective, verifying all aspects of the supply chain is a desire of the red meat industry. The geographical origin of feeds is of importance for particular brands or breeds (e.g. Wagyu) and section 3.3 provides insight into REIMS capacity for analysing plant products.

Lines 393-401: It will be very beneficial to show these data in summarizing table, if it is possible.

                Response: As the differences in fatty acid contents and odour and sensory between breeds are complex, especially where marbling levels are variable, it seems insufficient to summarise them in one single table. However, these differences have been well summarised across 3 separate tables in Frank (2016) and readers are encouraged to refer to this paper to view specific fatty acids or odour/ flavour attributes. To summarise in one table maybe misleading to the reader, as it is not any one particular fatty acid that is responsible for an odour or flavour note, but rather a combination of fatty acids and other volatiles/ non-volatiles that contribute.

Figure 4 -  It is illegible.

                Response: The resolution of Figure 4 (now Figure 1) has been enhanced and should now meet the requirements of the journal.

There are two font size (chapter 4 is written with smaller font size).

                Response: The font sizes used in the manuscript have been checked for consistency.

Reviewer 3 Report

The Authors have reviewed an interesting topic related to the  the REIMS technology, and its application to the areas of provenance, quality and safety to the red meat industry, particularly in an Australian context.

I would like to congratulate Authors for the good-quality of their article, the literature reported used to write the paper, and for the clear and appropriate structure. The manuscript is well written, presented and discussed, and understandable to a specialist readership.

In general, the organization and the structure of the article are satisfactory and in agreement with the journal instructions for authors.

The subject is adequate with the overall journal scope. The work shows a conscientious study in which a very exhaustive discussion of the literature available has been carried out.

The introduction provides sufficient background, and the other sections include results clearly presented and analyzed exhaustively.

As specific comment, with the aim to further improve the quality of the paper, an overall check of the English language is recommended.

Also, check the references style reported in Table 1.

The Conclusion section could be further improved.

The Authors are also invited to check if all references have been cited in the text as well as the citation format in the text.

So, in my opinion the paper merits the acceptance after minor revisions.

Author Response

Reviewer 3

The Authors have reviewed an interesting topic related to the REIMS technology, and its application to the areas of provenance, quality and safety to the red meat industry, particularly in an Australian context. I would like to congratulate Authors for the good-quality of their article, the literature reported used to write the paper, and for the clear and appropriate structure. The manuscript is well written, presented and discussed, and understandable to a specialist readership. In general, the organization and the structure of the article are satisfactory and in agreement with the journal instructions for authors. The subject is adequate with the overall journal scope. The work shows a conscientious study in which a very exhaustive discussion of the literature available has been carried out. The introduction provides sufficient background, and the other sections include results clearly presented and analyzed exhaustively.

            Response: We thank Reviewer 3 for providing comments on the manuscript and for acknowledging the clarity of the review and the overall strength of the information presented. Reviewer 3 also provided specific comments which we have addressed and our response to each item is listed below.

As specific comment, with the aim to further improve the quality of the paper, an overall check of the English language is recommended.

            Response: A further check of the language, spelling and readability has been conducted.

Also, check the references style reported in Table 1.

            Response: We thank the reviewer for bringing this to our attention. The reference style shown in Table 1 is now consistent with the journal guidelines.

The Conclusion section could be further improved.

            Response: The conclusion section has been modified slightly in response to specific comments made by the other reviewers. The authors do not agree that additional changes would add value to the review and believe that the summary section clearly captures the essence of the review and the potential utility of REIMS in the red meat industry.

The Authors are also invited to check if all references have been cited in the text as well as the citation format in the text.

            Response: All references have been cited in the text and the citation format in the text has been checked.

Round 2

Reviewer 2 Report

All my comments were  included in revised version of manuscript. I agree with publication in present form.